# Cumulative Risk and Externalizing Behaviors during Infancy in a Predominantly Latine Sample

**DOI:** 10.3390/bs13050363

**Published:** 2023-04-27

**Authors:** Greighson M. Rowe, Daniel M. Bagner, Nicole E. Lorenzo

**Affiliations:** 1Department of Psychology, American University, Washington, DC 20016, USA; 2Department of Psychology, Florida International University, Miami, FL 33199, USA

**Keywords:** parenting interventions, child externalizing behaviors, parenting skills, Latine

## Abstract

Externalizing behavior problems are among the top mental health concerns in early childhood, and many parenting interventions have been developed to address this issue. To better understand predictors of parenting intervention outcomes in high-risk families, this secondary data analysis evaluated the moderating effect of cumulative risk on child externalizing behaviors, parenting skills, and intervention dropout after completion of a home-based adaptation of the child-directed interaction phase of parent–child interaction therapy (PCIT) called the Infant Behavior Program (IBP). The participants included 58 toddlers (53% male; average age of 13.5 months; and 95% Hispanic or Latine) who were part of a larger randomized control trial in which families were randomly assigned to receive the IBP or treatment as usual (TAU). Cumulative risk was found to moderate the effect of the intervention group on child externalizing behaviors such that the participants in the intervention group with higher cumulative risk scores had greater reductions in externalizing behaviors. A potential explanation for these unexpected findings may be that the obstacles to treatment that were previously imposed by comorbid risk factors (i.e., lack of transportation, time commitment, and language barriers) were adequately addressed such that the families who most needed the intervention were able to remain fully engaged.

## 1. Introduction

Externalizing behavior problems in early childhood are a highly relevant public health concern, with disruptive behavior disorders being the most common reason for referral to child mental health clinics in the United States [1,2]. Common externalizing behavior disorders, such as attention-deficit/hyperactivity disorder and oppositional defiant disorder, have been associated with increased risks of adverse outcomes in academic achievement, comorbidity with other mental illnesses and substance abuse, and criminality [3]. In addition, populations with marginalized identities and systemic disadvantages are reported to have higher rates of disruptive behavior disorders and have higher likelihoods of comorbid psychosis, depression, and substance abuse [4]. The non-transient nature of infant–toddler behavioral problems highlights the importance of intervention as early as possible to effectively reduce the need for more intensive treatments later in life [5,6].

### 1.1. Parent–Child Interaction Therapy

A unifying theme across the theories of child development is the critical role of the primary caregiver. With this in mind, one of the most common methods for managing childhood behavioral problems is parenting interventions. Numerous parenting intervention programs have been developed that have demonstrated efficacy in the reduction of externalizing behaviors [7,8], with parent–child interaction therapy (PCIT) being one of the most highly researched and established evidence-based treatments for children who display disruptive behaviors as young as 3 years old [7]. The structure of the original intervention consists of weekly, hour-long, in-lab/clinic sessions across two phases of treatment. In the child-directed interaction (CDI) phase of treatment, parents are coached on improving the parent–child relationship during child-led play by increasing external displays of parental warmth and decreasing attention to child tantrums and negative behaviors. Therapists coach parents to use various *do* skills (i.e., praising appropriate behaviors, imitating appropriate play, etc.) and *don’t* skills, such as acknowledging inappropriate behavior, giving commands, asking questions, and being critical. To reach the skill criteria of the CDI-phase, caregivers must exhibit a sufficient number of *do* skills and a minimal number of *don’t* skills before moving on to the second phase of PCIT: parent-directed interaction (PDI). In the PDI phase, therapists teach caregivers to efficiently and appropriately give commands and increase child compliance, and treatment is considered complete once caregivers have achieved the skill criteria of both phases.

### 1.2. Dropout and Differential Responses to PCIT

Despite their effectiveness, the dropout rates of parenting programs are high [9,10], and sociodemographic risk factors increase the likelihood of a poorer response to PCIT [11]. A limitation of the traditional PCIT training model is the strict skill criteria, which can take a significant amount of time to meet. In one large study, 25% of families who graduated from PCIT took an average of 20.5 weekly sessions to achieve the skill criteria, with a range of 5 to 71 sessions [12]. In ethnic minority populations, it has been suggested that the timeline is even longer (i.e., 18 weeks longer) due to cultural differences in the adoption of the Western norms that PCIT tends to encourage [13,14], adding another layer of difficulty for families to achieve skill criteria status.

To increase the accessibility and effectiveness of PCIT and decrease dropout rates, many variations have been developed to reduce the time commitment and incorporate more culturally relevant methods. Adaptations are especially relevant given the demonstrated differences in skill acquisition and the logistical barriers of attending an indefinite number of sessions. For example, Spanish-speaking Latina mothers were found to use significantly more *don’t* skills than English-speaking Latina mothers post-treatment and during follow-up [15]. Additionally, Spanish-speaking PCIT therapists have been found to significantly differ in their total verbalizations and their use of directive and responsive coaching in comparison to English-speaking PCIT therapists [16,17]. One way to address the logistical barriers of completing traditional PCIT is by focusing treatment on the child-directed interaction (CDI) phase of PCIT, halving the required time commitment. The CDI phase alone has shown effectiveness in ethnically diverse and low socioeconomic status samples in the United States [14,18,19], as well as with Latine toddlers as young as 12 months with externalizing behavior concerns [20].

### 1.3. Individual Risk Factors of Intervention Response

Despite efforts to increase engagement with PCIT, various risk factors have been identified that reduce the effects of the intervention, many of which can be traced to systemic disadvantages, disproportionate stressors, and individual characteristics. For example, low socioeconomic status (SES) and maternal education are significant predictors of dropout from behavioral parenting interventions [21], and maternal intellectual functioning predicted 83 percent of families that dropped out of PCIT [13]. In addition to low SES, single-parent status and negative life stress predicted over 70 percent of poor parenting intervention outcomes one year after treatment [22,23]. Parental stress, in particular, has received recent attention due to its positive associations with child internalizing and externalizing behavioral concerns [24,25] and the interaction between mother–father stress and the risk of child abuse [26].

Identifying with a racial or ethnic minority status similarly places people at a higher risk for adverse outcomes. For example, Black youth are more likely to be diagnosed with externalizing behavioral disorders in comparison to White youth, who are more likely to be diagnosed with adjustment disorders [27]. Similar discriminant patterns are evident in the diagnosis of learning disabilities and other disorders relevant to child development in Black and Latine populations [28,29].

When considering disparities, it is also important to acknowledge that risk factors are related to one another. For example, low SES and ethnic minority status are more likely to co-occur in Hispanic and Black populations earning USD 12,000 and USD 22,000 less than the average median household income in the United States, respectively [30]. This finding aligns with prior research showing that low socioeconomic status (SES) and ethnic minority group membership predicted nearly 73% of *dropouts*, while low SES and White identification predicted 80% of treatment *completion* [21]. Language barriers also pose significant obstacles to treatment engagement for Latine populations seeking health care [31].

### 1.4. Cumulative Risk and Intervention Outcomes

Although most prior research examined the effect of each individual risk factor independently [18], there is evidence for better prediction of treatment outcomes and dropouts by combining individual risk factors of adverse treatment outcomes into one cumulative risk index. One such study found that participants with higher cumulative risk scores were significantly more likely to drop out from PCIT and be less responsive to treatment [11]. Another study on cumulative risk in parenting intervention outcomes did not find a significant relationship between cumulative risk and parent-reported problem behaviors after receiving the Family Check-Up intervention [32]. However, this study included children who were at risk for behavior problems, not those that were already exhibiting problematic behaviors. Additionally, no adaptations were made to improve accessibility and generalizability to diverse populations. Other studies have found a positive relationship between cumulative risk and child externalizing behavior such that higher levels of risk were associated with more externalizing behaviors [33], and multiple stressor groupings of 4 or higher predicted higher rates of externalizing behavior concerns [34].

### 1.5. Current Study

To date, an analysis of the effect of cumulative risk on treatment outcomes has not been conducted on a predominantly Latine toddler sample that received an in-home early behavioral parenting intervention with supportive adaptations to make care more accessible. The goal of the present study was to examine the moderating effect of cumulative demographic risk factors on child externalizing behaviors, parenting skills (i.e., *do* and *don’t* skills), and the likelihood of dropping out of the treatment. We hypothesized that (1) cumulative risk would moderate the relationship between the treatment group and child externalizing behaviors such that lower cumulative risk scores would predict fewer child externalizing behaviors in the intervention group, (2) cumulative risk would moderate the relationship between the intervention group and parenting skills such that lower cumulative risk scores would predict more *do* skills and fewer *don’t* skills in the intervention group, and (3) lower cumulative risk scores would decrease the likelihood of dropping out of the intervention.

## 2. Materials and Methods

### 2.1. Procedures

This study is a secondary data analysis of a larger randomized control trial in which mother–child dyads were randomly assigned to be in the Infant Behavior Program (IBP) or to receive treatment as usual (TAU) with their primary care provider. In contrast to traditional PCIT, the IBP was conducted in the participants’ homes to reduce transportation barriers to receiving the intervention. The program only included the CDI phase, was conducted in the language preferred by the family (English or Spanish), and was delivered as an early prevention program for at-risk toddlers. The IBP therapists were all clinical psychology doctoral students that were trained by the second author, a certified PCIT master trainer. Following the traditional PCIT protocol, the weekly sessions lasted approximately 60–90 min, with the first session being a teaching session in which therapists taught the families the *do* and *don’t* skills described above. The following sessions were coaching sessions that provided further support to families in implementing these skills. Families were directed to practice the skills for five minutes each day, and to log how frequently they used the skills at the end of each week. Families completed the intervention in an average of 6.1 sessions, with a range from 5 to 7 sessions, including the teaching session. Assessments were completed at baseline, immediately following the intervention, and at 3- and 6-month follow-ups. The main outcomes were reported elsewhere [20].

### 2.2. Participants

Recruitment was conducted at a pediatric primary care facility that serves predominantly underserved families. The study received approval from the institutional review boards of the University and Hospital and obtained informed consent from all participants prior to the screening procedures. The study’s inclusion criteria required children to score above the 75th percentile on the *Brief-Infant Toddler Social and Emotional Assessment*, a screener for problem behaviors in early childhood, as rated by their primary caregiver [35]. Additionally, mothers were required to score a 70 or higher on the Wechsler Abbreviated Scale of Intelligence (WASI), a widely used measure of general intelligence, or to score a 4 or higher on the Spanish-language equivalent, the *Escala de Inteligencia Wechsler para Adultos: Third Edition* [36,37]. Caregivers had the option of completing the study in either English or Spanish. For this study, 58 families were included in the analyses. The children were 53 percent male, had an average age of 13.5 months at baseline, and were 95% Hispanic or Latine. The toddler race demographics were as follows: 83% White, 7% other, 5% biracial, and less than 5% were American Indian/Alaska Native, Asian American, or Black. All primary caregivers were the children’s mothers, and 91% identified as Hispanic or Latina. The average age of the mothers was 29.57 years (SD = 5.49). In total, 83% of the mothers were White, 7% were Black, 7% identified as a race that was not listed, and less than 4% were Asian American or biracial.

### 2.3. Risk Measures

Informed by Bagner and Graziano’s (2013) cumulative risk index for behavioral parenting interventions, seven variables were identified as potential risk factors. The risk factors included socioeconomic status (annual income below the sample district’s federal poverty guidelines [38]), maternal education level (some college or less [21]), maternal intelligence (score of 79 or lower on the Wechsler Abbreviated Scale of Intelligence [13,37]), parenting stress (percentile score of 81 or higher on the Paternal Stress Index [24,25,39]), and marital status (single parent [22,40]). To account for familial ethnic or racial minority status as a risk factor, child ethnicity (identifying as Hispanic/Latinx) and race (identifying as non-White) were considered risk factors due to the diagnostic implications of ethnic and racial identity on children [27,28]. Each variable was then transformed into a dichotomous variable based on a predetermined cutoff score, with 1 indicating presence of risk and 0 indicating no presence of risk.

### 2.4. Outcome Measures

To assess child behavior, the *Infant-Toddler Social and Emotional Assessment* (ITSEA) was administered at baseline, immediately following the intervention, and at 3- and 6-month follow-ups. The measure is a 166-item questionnaire designed to assess child behavior in 12- and 36-month-olds. It has excellent test–retest reliability (*r* = 0.85 to 0.91) and good inter-rater reliability (*r* = 0.70 to 0.76 [35]). For the purpose of the current study, the externalizing domain, composed of the activity/impulsivity (6 items, α  = 0.22) and aggression/defiance (12 items, α  =  0.77) subscales, was used.

The *Dyadic Parent–Child Interaction Coding System-Third Edition (DPICS-III)* was used to code for observed parenting skills [41]. The DPICS has demonstrated psychometric validity for English- and Spanish-speaking mother–child dyads [14,20,42]. The frequency of mother *do* and *don’t* skills was evaluated during 5 min sessions of child-led play at all four data collection points. DPICS-III coders were blinded to the intervention groups and yielded an overall kappa of 0.84 [15].

To consider the intervention complete, participants must have attended the baseline session and the postintervention session. If either session was missed, the participants were classified as dropouts.

### 2.5. Statistical Plan

Data were analyzed using SPSS v. 28.0.1.1. To test the hypothesis of cumulative risk moderating the relationships between the intervention group and postintervention ITSEA externalizing scores and parenting skills, a multiple linear regression analysis was conducted using the PROCESS procedure for SPSS (version 4.2). Considering the high prevalence of ethnicity as a risk factor (94.8%), analyses were conducted both including and excluding ethnicity in the cumulative risk index, and they produced the same results. The analyses described are those including ethnicity due to the established importance of taking ethnicity into consideration when predicting externalizing behaviors, parenting skills, and treatment completion [15,21,27].

There were no significant differences in child sex, child age, or maternal age between groups, and child sex and baseline outcome scores were used as covariates for all regression analyses. The baseline predictor variables (cumulative risk, ITSEA externalizing scores, and *do* and *don’t* skills) were normally distributed and were centered during the PROCESS procedure analyses. All outcome variables showed linearity in quantile–quantile plots.

## 3. Results

### 3.1. Descriptive Analyses

The maximum possible number of risk factors was seven, and the numbers of participants for each cumulative risk score were 0 risks (*n* = 0), 1 risk (*n* = 1), 2 risks (*n* = 2), 3 risks (*n* = 12), 4 risks (*n* = 16), 5 risks (*n* = 16), 6 risks (*n* = 11), and 7 risks (*n* = 0). Table 1 describes the prevalence rates of each risk factor. The mean cumulative risk score was 4.33 (SD = 1.21) for the whole sample, 4.61 (SD = 1.27) for the TAU group, and 3.96 (SD = 1.02) for the IBP group. Appendix A Table A1 describes the mean outcome scores for each risk level. Fourteen percent (*n* = 8) of the families dropped out of the intervention.

### 3.2. Cumulative Risk and Child Externalizing Behaviors

A Pearson correlation coefficient was computed to assess the correlation between cumulative risk and postintervention ITSEA externalizing scores in the TAU group. As expected, there was a moderate positive correlation between the cumulative risk and externalizing behavior (*r* = 0.443, *p* = 0.006). Using a linear regression, cumulative risk predicted ITSEA externalizing scores in the TAU group (controlling for sex and baseline externalizing scores) (*F*(1, 29) = 7.063, *p* = 0.013, *R*^2^ = 0.196). Controlling for sex and baseline ITSEA externalizing scores, the overall model was significant (*F*(5, 42) = 10.28, *p* = 0.000, *R*^2^ = 0.550). As illustrated in Figure 1, the intervention group (IBP or TAU) and cumulative risk were found to significantly interact to predict ITSEA externalizing scores (*B* = −0.19, *p* = 0.016) such that for those in the IBP group, externalizing scores were assumed to decrease by −0.20 (*p* = 0.017) for cumulative risk scores of 4 and to decrease by −0.57 (*p* < 0.001) for cumulative risk scores of 6. This analysis was repeated with the activity/impulsivity subscale (*B* = −0.02, *p* = 0.794) and the aggression/defiance subscale (*B* = −0.06, *p* = 0.500), but no significant interaction effects were found. The analysis was also repeated for the 3-month (*B* = −0.03, *p* = 0.715) and 6-month (*B* = −0.08, *p* = 0.376) follow-ups, but no significant interaction effects were found.

### 3.3. Cumulative Risk and Parenting Skills

For the parenting skills analyses, the intervention language (English or Spanish) was included as a covariate in addition to sex and baseline parenting skills due to prior findings of language differences in the utilization of do and don’t skills [15]. The overall model evaluating the moderating effect of cumulative risk on the association between the intervention group and the change in do skills was not significant (F(6, 38) = 2.23, *p* = 0.061, *R*^2^ = 0.26). The analysis was repeated for the 3-month (*B* = −2.29, *p* = 0.437) and 6-month (*B* = −2.96, *p* = 0.368) follow-ups, and no significant interaction effects were found. The overall model evaluating the moderating effect of cumulative risk on the relationship between the intervention group and the change in don’t skills was significant (F(6, 38) = 8.61, *p* < 0.001, *R*^2^ = 0.576). As illustrated in Figure 2, the intervention group (IBP or TAU) and cumulative risk were found to significantly interact to predict don’t skills (*B* = 5.00, *p* = 0.034). For those in the TAU group, don’t skill frequency significantly decreased by −12.55 (*p* = 0.004) for cumulative risk scores of 3. The analysis was repeated for the 3-month (*B* = 3.10, *p* = 0.260) and 6-month (*B* = −2.14, *p* = 0.435) follow-ups, but no significant interaction effects were found.

### 3.4. Cumulative Risk and Dropout

A binary logistical regression analysis was conducted to evaluate the relation between cumulative risk and dropout in the IBP group. The overall model was significant (*B* = 0.887, *p* = 0.048), but cumulative risk was not a significant predictor of intervention dropout (χ^2^ = 0.000, *p* = 1.00).

## 4. Discussion

This study evaluated the moderating effect of cumulative risk factors on various outcomes of the Infant Behavior Program in children between 12 and 15 months old. It was expected that (1) cumulative risk would moderate the relationship between the treatment group and child externalizing behaviors such that lower cumulative risk scores would predict fewer child externalizing behaviors in the intervention group, (2) cumulative risk would moderate the relationship between the intervention group and parenting skills such that lower cumulative risk scores would predict more *do* skills and fewer *don’t* skills in the intervention group, and (3) lower cumulative risk scores would decrease the likelihood of dropping out of the intervention. The results challenged some of the hypotheses and supported others on the effects of cumulative risk on child behavior, parenting, and intervention dropout.

### 4.1. Child Externalizing Behaviors

Consistent with prior research on cumulative risk, the results demonstrated that high cumulative risk scores were associated with increases in externalizing behaviors in the TAU group [33,34]. However, higher cumulative risk scores predicted a reduction in externalizing behaviors in the IBP group such that those who were assessed to have greater risk had better externalizing behavioral outcomes. Although Gardner et al. [32] did not find that the cumulative risk index significantly predicted problem behaviors in a family-centered intervention, as found in the present study, they found that low parental education levels predicted greater responsiveness to the intervention, similarly finding the greatest success when in the presence of a certain risk factor. A potential explanation for these unexpected findings may be that the obstacles to treatment that were previously imposed by comorbid risk factors (e.g., lack of transportation, time commitment, and language barriers) were adequately addressed such that the families who most needed the intervention were able to remain fully engaged. The significance of the externalizing behavior outcomes also likely overlaps with the relatively low dropout rate for a high-risk sample (15.8%) and the lack of an association between risk and dropout, as discussed below. Cultural components in response to parenting interventions should also be explored, with Latine parents reporting a stronger desire to engage in parenting interventions that are respectful of norms between cultures and demonstrate an understanding of their lived experiences [43].

### 4.2. Parenting Skills

Although the moderating effect of cumulative risk on *do* skills was not significant, a significant interaction effect was found for *don’t* skills such that those in the treatment as usual group with cumulative risk scores of 3 showed a significant decrease in *don’t* skill frequency. A study using the same dataset similarly found significant differences in the use of *don’t* skills in Spanish-speaking versus English-speaking mother–child dyads but no significant difference in the use of *do* skills [15]. This may be attributable to the maternal demographic risk characteristics, as it is consistent with previous research where mothers with lower levels of education had fewer overall verbalizations when playing with their children, accounting for the non-significant effect on *do* skills [44,45]. The percentage of mothers who had completed some college or less in our sample was 69% (*n* = 40), which could account for a considerable amount of the variability in skill implementation.

### 4.3. Dropout

Additionally, despite risk factors for dropout being a major concern in prior studies of parenting interventions [19,21], cumulative risk was not found to be significantly correlated with dropout in this sample. These results contrast with the study that inspired the model of cumulative risk used for this sample [11]. This difference may be due to the difference in the sample sizes for each risk factor between the two samples, with Bagner and Graziano’s sample having 15 dyads with a risk score ≥ 3, and the present sample having the majority of dyads (95%) with a risk score ≥ 3.

Another potential reason for risk not predicting dropout may be the numerous adaptations made to the Infant Behavior Program in comparison to traditional PCIT. Due to the session-limited nature of the intervention, the elimination of language barriers, and the availability of in-home sessions, several significant obstacles to retention were accounted for from the onset of the intervention. These practical and cultural adaptations align with previous research on the efficacy of modifications to PCIT [18,20,42].

### 4.4. Limitations and Future Research

In interpreting these findings, one limitation of the study is the relatively small sample size. Moreover, the comparison group was standard primary care, rather than a comparable intervention. Thus, it cannot be stated for certain that the unique features of the Infant Behavior Program are superior to alternative parenting interventions for families with toddlers. Additionally, significant results were not found for the 3- and 6-month follow-up outcomes. Future research should continue to explore the efficacy of early prevention programs in diverse child populations, including the various adaptations that have already been developed to increase the accessibility of PCIT interventions via telehealth [46,47].

## 5. Conclusions

By acknowledging the high comorbidity of risk factors when predicting intervention outcomes and addressing treatment barriers, this study showed the potential clinical utility of using cumulative risk to assess variance in externalizing behavioral changes, parenting skills, and dropout in parenting interventions. Additionally, these findings provide evidence for the value of modifying interventions to the needs of the sample population, highlighting the need to target those who are at the highest risk, as they can have the largest gains with small changes to accessibility.

## Figures and Tables

**Figure 1 behavsci-13-00363-f001:**
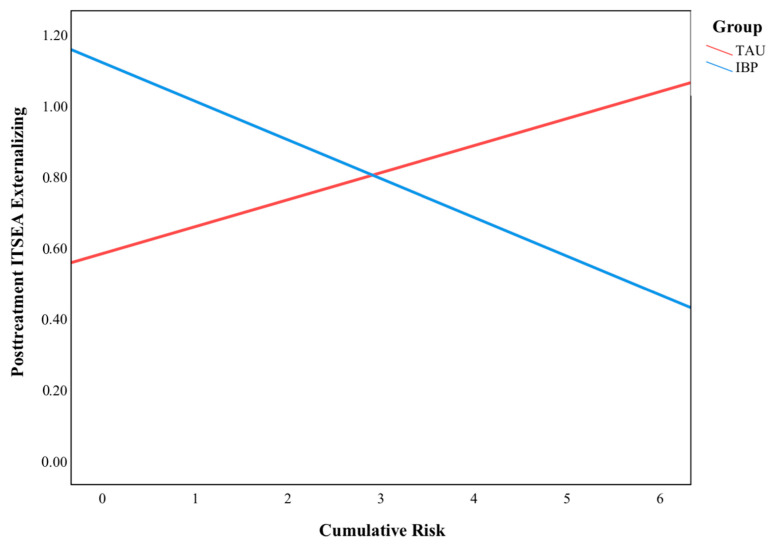
The moderating effect of cumulative risk on ITSEA postintervention externalizing behaviors.

**Figure 2 behavsci-13-00363-f002:**
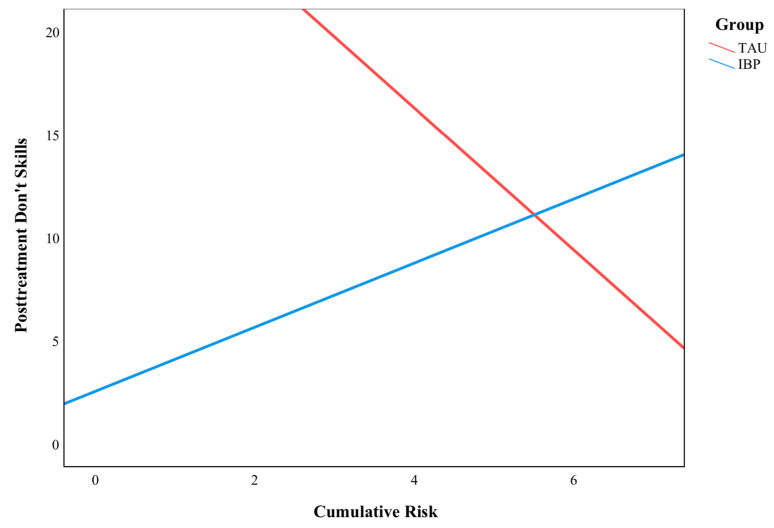
The moderating effect of cumulative risk on don’t skills.

**Table 1 behavsci-13-00363-t001:** Prevalence of risk factors.

Measures	% Risk Factor
SES	87.9
Maternal Education	69.0
Maternal Intelligence	82.8
Marital Status	31.0
Parenting Stress	50.0
Child Ethnicity	94.8
Child Race	17.2

## Data Availability

No new data were created or analyzed in this study. The data presented in this study are available on request from the second author.

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
