# Peer review of "Cumulative Risk and Externalizing Behaviors during Infancy in a Predominantly Latine Sample"

_behavsci, 2023, doi:10.3390/bs13050363_

Round 1

Reviewer 1 Report

Comments Reviewer.

This reviewer considers it relevant to evaluate interventions after carrying out an intervention programme, an aspect that is often not carried out. Hence the relevance of the study. In addition, the adaptations made to the programme and the intervention in the context of the home should be highlighted.

Some aspects that this reviewer raises after reading the work are the following:

Nothing is specified about the ethics committee, participation consents.....

The procedure could include more details related to the professionals who carried out the intervention, such as years of training experience in this programme, for example. As well as the final number of sessions and the specific objectives of each session.

In the description of the sample I would include the age of the mothers. This is a risk factor if we are talking about teenage or premature mothers. In addition, it is to be expected that there are no differences in age between the two groups, although this is not mentioned in the article.

Two figures appear in the manuscript, although they are not referenced in the text.

Reviewer 2 Report

This is a well written and structured article and makes a particularly interesting point about the language preference of the therapist. The results and conclusion are novel.

I have one minor comment about line 286: needs clarification, I assumed this was reference to parents 'some or less' education? but I could be wrong, so please clarify.

Reviewer 3 Report

Dear Authors,

Thank you very much for the opportunity to revise this work on the predictors of parenting intervention outcomes in high-risk families.

I believe this is an important work with several implications for clinical practice. However, some concerns should be addressed before the official publication in Behavioral Science Journal.

  • Before the presentation of the results, I think it would be beneficial to add a statistical plan paragraph in order to guide the readers through the statistical analyses that had been conducted. First of all, were the data checked for normality and linearity?

I would suggest the author provide more information.

  • In general, the discussion part should be deepened and further discussed, given also the presence of several unexpected results
